# Analysis of Deformation Characteristics of Foundation-Pit Excavation and Circular Wall

**Xuhe Gao [1],\*** , **Wei-ping Tian [1] and Zhipei Zhang [2]**

[1] Key Laboratory of Highway Engineering in Special Region, Ministry of Education, Chang'an University, Xi'an 710064, China; fz02@gl.chd.edu.cn

[2] College of Geology and Environment, Xi'an University of Science and Technology, Xi'an 710054, China; zhipeizhangt@126.com

\* Correspondence: 2017021008@chd.edu.cn

**Abstract:** The surrounding ground settlement and displacement control of an underground diaphragm wall during the excavation of a foundation pit are the main challenges for engineering safety. These factors are also an obstacle to the controllable and sustainable development of foundation-pit projects. In this study, monitoring data were analyzed to identify the deformation law and other characteristics of the support structure. A three-dimensional numerical simulation of the foundation-pit excavation process was performed in Midas/GTS NX. To overcome the theoretical shortcomings of parameter selection for finite-element simulation, a key data self-verification method was used. Results showed that the settlement of the surface surrounding the circular underground continuous wall was mainly affected by the depth of the foundation-pit excavation. In addition, wall deformation for each working condition showed linearity with clear staged characteristics. In particular, the deformation curve had obvious inflection points, most of which were located deeper than 2/3 of the overall excavation depth. The characteristics of the cantilever pile were not obvious in Working Conditions 3–9, but the distribution of the wall body offset in a D-shaped curve was evident. Deviation between the monitoring value of the maximal wall offset and the simulated value was only 4.31 %. The appropriate physical and mechanical parameters for key data self-verification were proposed. The concept of the circular-wall offset inflection point is proposed to determine the distribution of inflection-point positions and offset curves. The method provides new opportunities for the safety control and sustainable research of foundation-pit excavations.

**Keywords:** circular foundation pit; construction monitoring; numerical simulation; underground continuous wall

## 1. Introduction

Underground continuous walls have been widely applied as foundation-pit supports due to their high stability, rigidity, and impermeability, in addition to their predictable deformation characteristics. However, for circular anchor foundation pits with underground continuous walls as the predominant retaining structure, monitoring and predicting wall displacement and surface settlement around the foundation pit remain challenges. As such, these factors need to promptly and consistently be monitored, and monitoring data should be accordingly analyzed. Challenges have also inspired scholars to explore new research methods, promoting the application of computer technology in the construction of foundation pits.

Studies on this topic have been conducted. Bolton and Powriet [1] carried out various laboratory tests to study the deformation characteristics of an underground continuous wall under different soil conditions and foundation-pit parameters. They also calculated the deformation and failure

conditions of the foundation pit. However, they did not discuss the validity of the used parameters in the calculation. Pohetal. [2] collated the monitoring data of two foundation pits, and used real-world data to calculate the bending moment generated by the underground continuous wall. Results showed that the bending moment of the underground continuous wall was largely generated due to the cracking of the wall, and that the lateral displacement of the wall was not affected by this factor. However, the study did not provide any description of the monitoring-data collection, nor did it further demonstrate the factors and characteristics of the lateral displacement of the wall. Bose and Som [3] created a more accurate finite-element-analysis program based on the Cambridge model to address deficiencies of the existing model, which is mainly used for calculations and analysis of the internal supports of a foundation pit. However, that study also lacked a demonstration of the validity of the model parameters. Whittle et al. [4] innovatively integrated two-dimensional seepage into the deep-foundation-pit calculation model, and examined soil stress in the deep-foundation-pit engineering of a postal building in Boston on the basis of the finite-element method. However, the study did not discuss the displacement and surrounding settlement of the support structure during excavation of the foundation pit. To better analyze foundation-pit support systems, Kishnani and Borja [5] conducted detailed analysis of the soil structure and seepage into the foundation pit, and analyzed the impact of these two factors on the support system. They determined that the seepage affected earth pressure behind the wall and caused the surrounding ground to settle. The effect of seepage on wall displacement, however, was not discussed. After summarizing multiple theories and practical experiences, Alejano et al. [6] conducted a related investigation on the factors affecting the displacement of typical structural types (filling and excavation). Soil traits were regarded as ideally elastoplastic, and it was noted that the displacement of the soil was not the only factor; the physical properties of the soil and the wall, as well as the location of the erected supports also contributed. That study did not involve ground settlement around the foundation pit, and the quantitative analysis of wall displacement was insufficient. Faheem et al. [7] focused on the poor stability of foundation pits in areas with soft soil from a two- and a three-dimensional perspective. Their study particularly focused on the stability of the bottom of the pit, and presented a detailed simulation using the finite-element method. However, there was no analysis of ground settlement around the foundation pit and the deformation of the supporting structure, and the validity of the parameters in the simulation process was not verified.

Liu and Ding [8] used the finite-element method to study the stiffness coefficient of the Goodman unit, which was determined to affect surface settlement outside the foundation pit and the displacement of the underground continuous wall. That study also failed to verify the validity of the finite-element calculation parameters. Chen et al. [9] investigated the deep foundation pit of a steel plant in Shanghai on the basis of collected monitoring data during foundation-pit construction. They analyzed the deformation and internal structural forces of the circular underground continuous wall supporting the foundation pit that was subjected to the pressure of confined groundwater. The study focused on analysis of existing monitoring data, and did not use finite-element analysis to further demonstrate the deformation characteristics of the supporting structure. Xu et al. [10] collected monitoring data from foundation pits in Shanghai that used underground continuous wall supports, and calculated the deformation law of the underground continuous wall to study the influence of various factors on these laws. Their study focused on regional data collected by statistical analysis, and had limited applicability to early warnings on surrounding surface settlement and wall displacement in special geological environments. Wang and Hu [11] studied the double-layer elliptical supporting structure in the foundation pit of the China Petroleum Building, and aimed to reduce the number of layers supported by the internal structure during the excavation of the foundation pit. The structure was analyzed by force-deformation calculations. It was concluded that a T- or I-shaped underground continuous wall could be used instead of the elliptical wall shape, which could reduce the number of required internal supports. That study lacked monitoring data or finite-element simulation to validate the results, and there was no analysis of surface settlement and wall displacement around

the foundation pit. Hu et al. [12] used the foundation pit of a subway station as a research subject, and monitored the variation of the horizontal displacement of the underground continuous wall at the excavation depth during the construction of the foundation pit. A three-dimensional finite-element model was established to simulate the foundation-pit excavation of the subway station, and the calculated deformation characteristics were compared with the monitoring results. Results showed that the difference between the simulated maximal horizontal displacement of the underground continuous wall and the measured value was small, and that the trend in displacement was comparable. However, the study also failed to verify the validity of the finite-element calculation parameters, and did not analyze ground settlement around the foundation pit. Zheng et al. [13] used finite-difference software FLAC3D to numerically simulate the horizontal deformation and surface settlement of a foundation-pit-excavation support structure and compared it with the measured values. Results showed that the maximal horizontal displacement of the underground continuous wall appeared at the top of the wall, and the horizontal displacement curve exhibited a "half-cup" composite shape with multiple inflection points. The settlement curve of the ground surface beyond the wall was an asymmetrical groove-type curve. Similarly, that study verified the validity of the finite-element calculation parameters.

In summary, the existing literature has conducted a large number of theoretical calculations and finite-element analysis of underground continuous walls (including self-programming and commercial software). However, there is almost no argument concerning the method of obtaining parameters. This shows that the method of parameter selection needs further study. If only research results are pursued, and access to key parameters is ignored, such research is questioned by other disciplines, and the sustainability of that work is also threatened. In this study, the anchored circular underground continuous wall of the Humen Second Bridge West foundation-pit project was monitored and simulated. Monitoring data were analyzed to identify the deformation law and other characteristics of the support structure. Three-dimensional numerical simulation of the foundation-pit excavation was conducted in Midas/GTS NX. To overcome the theoretical shortcomings of parameter selection for finite-element simulation, the key data self-verification method was used, and a layer-by-layer algorithm was employed to determine more accurate simulation parameters. The deviation rate was used to quantify the difference between simulated results and measured values. The appropriate physical and mechanical parameters for key data self-verification were proposed and utilized to compensate for the shortcomings of the on-site monitoring data. The concept of the "circular-wall offset inflection point" was proposed to determine the distribution of inflection-point positions and offset curves. The method provides new opportunities for the safety control and sustainable research of foundation-pit excavations.

## 2. Materials and Methods

### 2.1. Project Overview

The rock and soil layers in the foundation pit were silt, muddy soil, fine sand, medium sand, coarse sand, strong weathered mudstone, middle weathered mudstone, and microweathered mudstone (Figures 1 and 2). According to these geological conditions and the design requirements of the anchor body, the underground continuous wall adopted a circular structure with an outer diameter of 82.0 m and a wall thickness of 1.5 m. The elevation of the top surface of the pit was 1.00 m, and the elevation of the bottom of the pit was −35.00 to −43.00 m. The bottom of the pit was embedded in mud, siltstone, and moderately weathered mudstone strata. The underground continuous wall was divided into two sections (Sections 1 and 2). Section 1 was three-milled, with a side groove length of 2.8 m, a middle slot length of 1.47 m, and a slot length of 7.07 m; Section 2 had a slot length of 2.8 m. The length of the Sections 1 and 2 groove sections was 0.25 m on the axis of the ground wall, and Sections 1 and 2 had 27 slots. Thus, the trough section was divided into 54 sections (Figure 2). The designed maximal trough depth was 46.0 m. On both sides of the underground continuous wall, a 50 cm diameter

cement-powder spray was used to create a pile to reinforce the silt soil with a spacing of 40 cm and a reinforcement depth of 15.0 m. After construction of the underground continuous wall was completed, the bottom of the wall was grouted.

After construction of the underground continuous wall had been completed, the soil was excavated by the reverse method, and the lining of the pit was layered and constructed. The construction period of each layer was controlled by the excavation of the soil. The excavation depth of the soil was 27 m, and the lining and soil-stratification height were controlled within 3 m. The lining of the pit was constructed from top to bottom. The top and bottom plates were 6 m thick with a concrete-filled core in the middle.

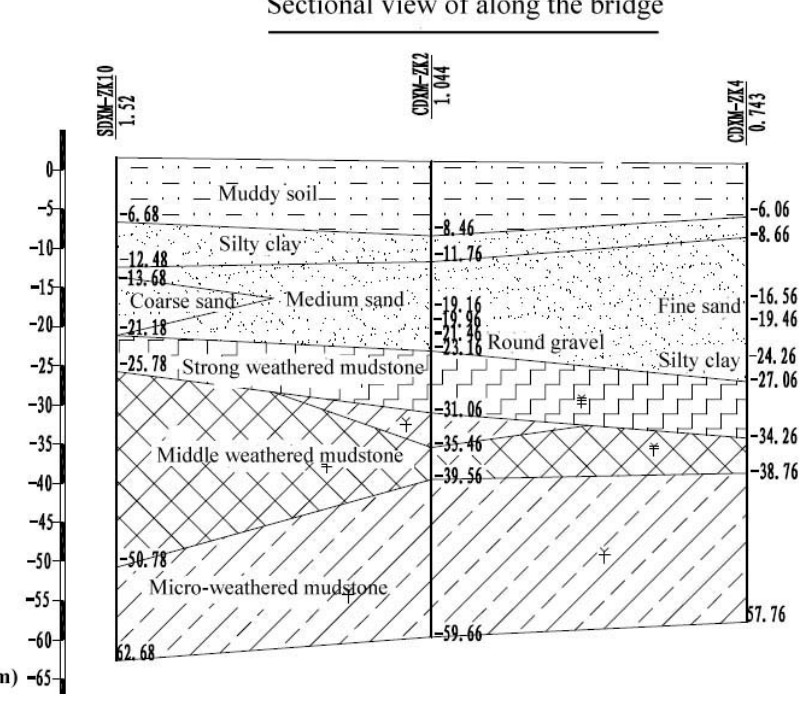

**Figure 1.** Cross-sectional view of geological section along the bridge.

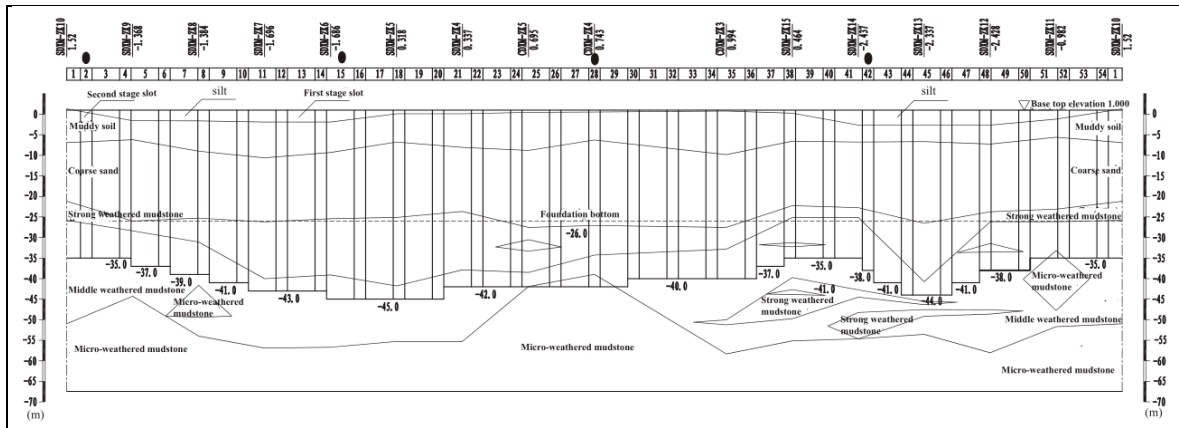

**Figure 2.** Expanded view of slots.

### 2.2. Surface-Deformation Monitoring around Underground Continuous Wall

Because of the need for surface-settlement monitoring during construction, a group of sensors were arranged to the east, south, west, north, southeast, northeast, southwest, and northwest of the foundation pit. Typical settlement monitoring started from the outside of the foundation pit with 10 monitoring points arranged at equal intervals of 5 m numbered D1-i to D8-i (with i = 1–10). Due to on-site construction-monitoring points that were actually available, only the first five points of valid data were obtained. A total of eight settlement-monitoring sections and 80 surface-settlement monitoring points were set. If the points encountered obstacles, they could be moved in parallel, as shown in Figure 3.

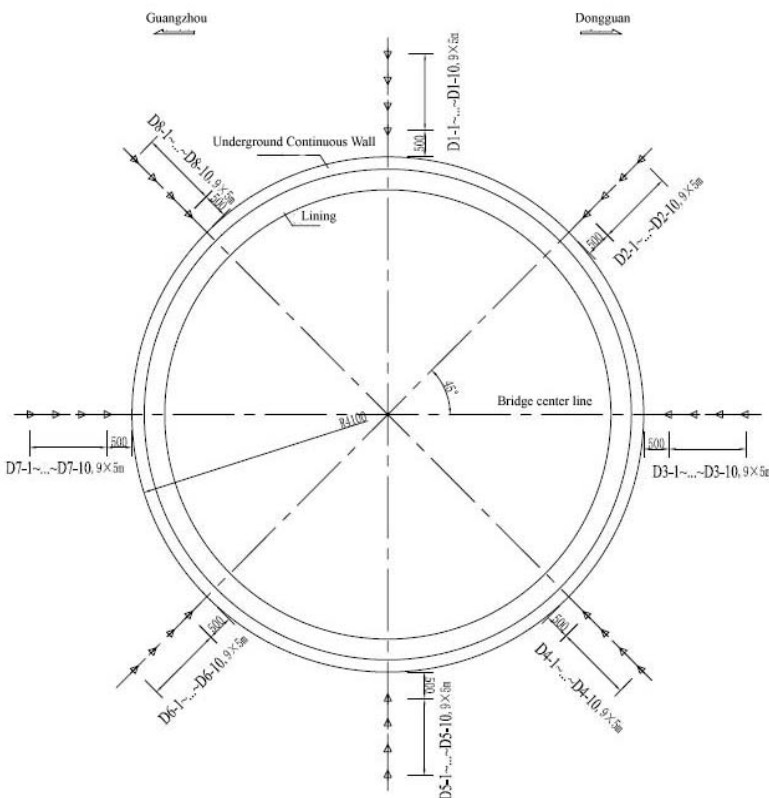

**Figure 3.** Layout of surface-settlement monitoring sites.

### 2.3. Deep-Lateral-Deformation Monitoring of Underground Continuous Wall

Deep-lateral-deformation monitoring of the underground continuous wall is a key component of monitoring and measuring the deformation of the foundation-pit support, which can directly reflect the safety and stability of the foundation pit and its supporting structures (Figure 4). To ensure the effective functioning of the inclined pipe fitting under the effects of high-pressure concrete, the inclined measuring holes were repeatedly arranged according to the spare hole position in the groove section where the four inclined measuring pipe parts of P1, P3, P5, and P7 were located (P1', P3', P5', P7'). There were a total of 12 inclinometer tubes.

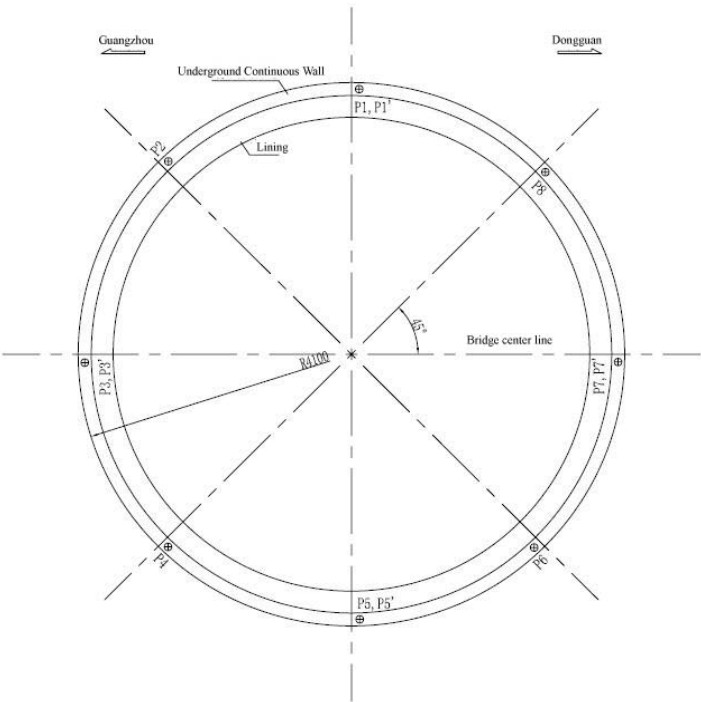

**Figure 4.** Deep-deformation monitoring site layout for underground continuous wall.

## 2.4. Monitoring-Data Analysis

The underground continuous wall was divided into 54 slot segments for analysis, as shown in Figure 2. To facilitate the statistical data processing, surrounding-settlement and wall-offset data corresponding to slot segments 2, 15, 28, and 42 were selected. In working-condition simulations, these four slot segments were defined to correspond to the calculation results of the four diagonal directions of the model.

## 2.5. Mohr–Coulomb Strength Criterion

The Mohr–Coulomb strength criterion states that shear failure is the most fundamental cause of soil failure. The shear strength of any point in the soil is only related to normal stress $\sigma_n$ on the plane, such that

$$\tau_f = f(\sigma_n). \tag{1}$$

This function is a curve in $\tau_f$-$\sigma$ co-ordinates, known as the molar-intensity line. The Moore envelope can be approximated as a linear relationship, known as the Coulomb equation, as follows:

$$\tau_f = c + \sigma_n \tan \phi, \tag{2}$$

where $\tau_f$ is the shear strength at any point in the soil, and $\sigma_n$ is the normal stress on the calculated plane.

Equation (3) is the stress condition at any point in the soil under the limit equilibrium state (stress is positive with compression). This equation is known as the Mohr–Coulomb strength criterion. The radius of the stress Mohr circle is

$$R = \left( \frac{c}{\tan \phi} + \frac{\sigma_{11} + \sigma_{22}}{2} \right) \sin \phi = c \cos \phi + \frac{\sigma_{11} + \sigma_{22}}{2} \sin \phi, \tag{3}$$

where $\sigma_{11}$ and $\sigma_{22}$ are the maximal and minimal principal stress when the plane-soil mass under goes shear failure, respectively.

When the Mohr envelope is tangential to the most stressed Mohr circle in the material, the soil undergoes shear failure. In other words, the magnitude of $\sigma_{22}$ has no effect on shear strength. The Mohr–Coulomb strength criterion is an irregular hexagonal cone in the principal stress space. The projection of the hexagonal cone on the $\pi$ plane is an irregular hexagon.

The Mohr–Coulomb criterion is widely used, as the constitutive model can accurately reflect the unequal tensile and compressive characteristics of geotechnical materials. However, numerical calculations for this model are prone to nonconvergence due to the discontinuous corners of the hexagonal cone.

### 2.6. Establishing Model of Foundation-Pit Excavation

Since the classical yield criterion ignores the frictional component of soil shear strength, such criteria can be used for the undrained analysis of saturated soils, such that $\varphi = 0$. The Mohr–Coulomb criterion surpasses classical criteria and considers the frictional component of the soil, which is more suitable for most scientific research and engineering practice. It is also more widely used in numerical simulation. Finite-element software Midas/GTS NX was used for numerical simulation analysis on the basis of the Mohr–Coulomb constitutive model.

The excavation project described in this study included a two-part supporting structure consisting of the underground continuous wall and the lining. The lower end of the underground continuous wall was embedded in the middle weathered-rock layer, and the embedded depth range was 10–20 m. In numerical simulation, it is necessary to simplify the foundation-pit excavation support model and the construction steps to ensure computational capacity and accuracy. The underground continuous wall retaining structure was constructed before the foundation pit was excavated. The excavation method selected a single layer of flat excavation and added a layer lining after the excavation of each layer was completed. This process continued until all construction steps were performed.

The soil layers are described in Section 2.1. Each soil layer was distinguished by a natural planar interface. According to the construction conditions and the topography of the project, the top surface of the calculation model was selected as the ground and defined as a free surface. The four lateral sides of the design model were also defined to limit horizontal displacement. The bottom plane of the pit was defined to limit vertical displacement. The initial self-weight stress field was the main model load condition. The design calculation model used the Mohr–Coulomb elastoplastic strength criterion. In addition, the river levee was approximately 50 m away from the foundation pit. In the numerical-calculation process, this levee was considered according to the most unfavorable situation for the excavation project.

The size of the design-calculation model was carefully selected to be 300 m long, 300 m wide, and 100 m deep. Errors in slot sections at segmentation were caused by errors on the construction site. The channel sections neighboring certain modelling errors were collected by overlap. Thus, the thickness of the simplified underground continuous wall was calculated as 1.3 m. The model was divided into various sections (Figures 5 and 6).

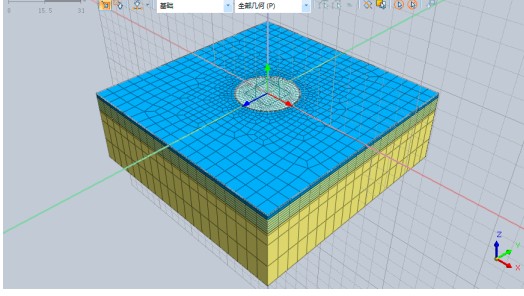

**Figure 5.** Pit-model grid diagram.

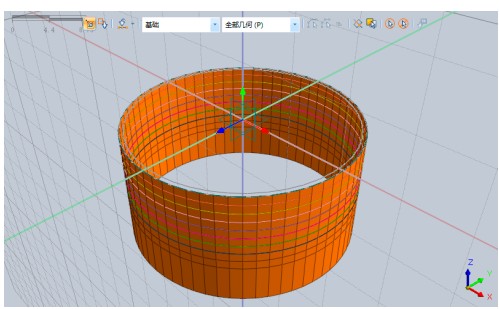

**Figure 6.** Support-structure grid diagram.

The model had a total of 15,840 units and 17,680 nodes. The first layer in the model was a silt layer with a thickness of 2 m; the second layer was a silty-clay layer with a thickness of 5 m; the third, fourth, and fifth layers of silt, and the medium and coarse-sand layers had a thickness of 6 m; the sixth, seventh, and eighth layers were strongly weathered mudstone, moderately weathered rock, and slightly weathered rock layers, with thicknesses of 15, 30, and 30 m, respectively.

The thickness of the underground continuous wall was calculated as 1.3 m. The thickness of the inner lining was 1.5 m in the range of 0–6 m depth, and thickness was 2 m below 6 m depth.

### 2.6.1. Selection of Physical and Mechanical Parameters

In the finite-element model, the parameters of the concrete material were assigned according to the defined specifications. The mechanical parameters of the rock layer were determined by geotechnical testing and the key data self-validation method. The required physical and mechanical parameters to calculate the constitutive equations in the model are shown in Tables 1 and 2.

**Table 1.** Soil parameters and indices.

| Soil Layer | Elastic Modulus (kN/ m$^2$) | Poisson Ratio | Angle of Internal Friction (°) | Cohesive Forces (kN/m$^2$) | Unit Weight (kN/m$^3$) |
|---|---|---|---|---|---|
| Silt | 3000 | 0.30 | 3 | 5 | 15.40 |
| Muddy soil | 50,000 | 0.27 | 5 | 8 | 16.50 |
| Fine sand | 80,000 | 0.23 | 18 | 0 | 19.00 |
| Medium sand | 120,000 | 0.24 | 25 | 0 | 19.50 |
| Coarse sand | 200,000 | 0.22 | 28 | 0 | 18.80 |
| Strong weathered mudstone | 500,000 | 0.19 | 20 | 50 | 19.99 |
| Middle weathered mudstone | 1,000,000 | 0.17 | 30 | 450 | 20.50 |
| Microweathered mudstone | 1,400,000 | 0.15 | 35 | 600 | 20.70 |

**Table 2.** Structural and mechanical parameters.

| Structure | Elastic Modulus (kN/ m$^2$) | Unit Weight (kN/ m$^3$) | Poisson Ratio |
|---|---|---|---|
| Underground Continuous Wall | $3.0 \times 10^7$ | 25 | 0.2 |
| Lining | $3.0 \times 10^7$ | 25 | 0.2 |

Note: Elastic modulus: ratio of stress to corresponding strain when ideal material has small deformation. Poisson ratio: ratio of absolute value of transverse normal strain to axial normal strain when material is under uniaxial tension or compression. Angle of internal friction: friction characteristics caused by mutual movement of particles and gluing. Cohesive forces: mutual attraction between adjacent parts within same substance. Unit weight: gravity characteristic of an object due to its gravitation in the natural state.

### 2.6.2. Calculation Process for Excavation-Pit Model

According to the support and excavation process for the circular-underground-continuous-wall foundation pit, pit simulations were calculated and analyzed for nine working conditions. Specifically, steps shown in Figure 7 were performed.

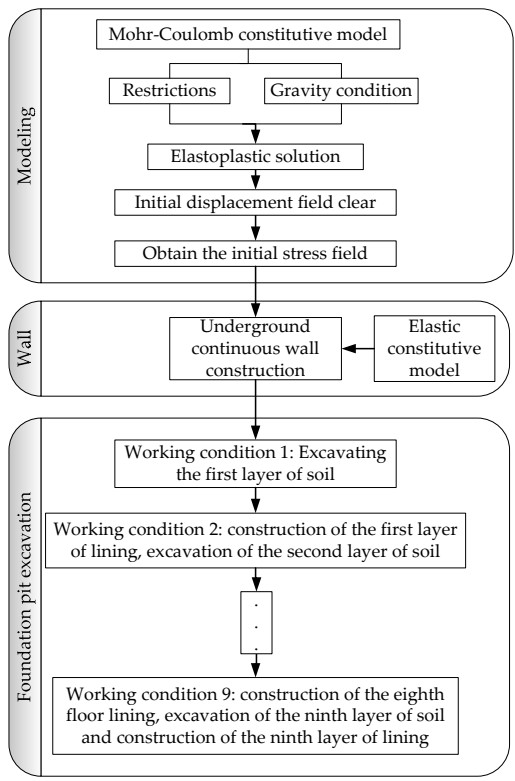

**Figure 7.** Modeling and calculation workflow.

### 2.6.3. Key Data Self-validation and Divisional-Condition Calculations

Stability analysis and the quantitative calculation of the supporting structure of existing foundation-pit engineering are mainly controlled by several key geotechnical parameters. The determination of parameters has always been a matter of debate in this field. Current practices are ① obtained by geotechnical tests, ② based on statistical data obtained from a large number of similar strata, and ③ empirical data. Because obtained parameters by geotechnical tests are different from the actual project, they need to be corrected. The method of statistical data is only applicable to ordinary strata and requires a lot of construction accumulation. Empirical data are easy to use, but are obviously less scientific. In addition, the three existing parameter-acquisition methods have a fatal disadvantage for engineering special geological environments: the parameter-selection method is not universal, and it is less sustainable.

Therefore, for traditional theoretical calculations and finite-element analysis, obtaining a method that could self-verify key data on the basis of project-site-monitoring data is critical to the sustainable development of foundation-pit and geotechnical engineering.

This paper proposes a key data self-validation theory. More specifically, we propose the selection of physical and mechanical parameters for numerical simulation that should be as reasonable as possible. However, due to many potential sources of uncertainty in these values, including theoretical defects that simplify soil and rock into ideal homogeneous materials, acquisition processing, and data conversion, when the parameter-selection basis was not sufficiently convincing, the key data obtained by monitoring were used to verify the results obtained by the simulation. When the deviation rate of the data obtained by the simulation was within a reasonable error range, the physical and mechanical parameters selected for the calculation model were deemed reasonable. Following this, large-scale data calculations were performed.

This method requires trial calculation. During research, parameters obtained from the literature and background data were used for trial calculations, and we calculated the deviation rate multiple times. Finally, the maximal simulated offsets of the walls under the second and third working conditions

were 1.31 and 2.25 mm, respectively. The maximal monitored offsets of the wall for the second and third working conditions were 1.58 and 2.57 mm, respectively. That is, the difference between the simulated and monitored values was calculated. The absolute value of the difference divided by the monitoring value was used to quantify the credibility of the simulated value. It was further verified that the parameters used in the simulation were feasible. The deviation rate was calculated as follows:

$$\text{Deviation rate} = (\text{analog value} - \text{monitored value})/\text{monitored value}. \qquad (4)$$

On the basis of this equation, the deviation rate of the wall was −10.39% for the second working condition and −14.22% for the third working condition. Thus, the obtained data from the simulation demonstrated a limited deviation, and the preliminary verification data were valid.

After having determined the appropriate parameters, the input parameters were calculated to obtain the force-deformation characteristics of other conditions. The calculation results were confirmed by result monitoring. Another benefit of this method is that it could expand the scope of the simulation calculations to compensate for the lack of on-site monitoring data.

## 3. Results

### 3.1. Surface-Settlement-Monitoring Analysis

The maximal settlement value of the monitoring points was 9.9 mm. For excavation Working Conditions 1–3, surface settlement at each monitoring point increased linearly. For Working Conditions 4–6, the monitoring points generated relatively stable settlement. For Working Conditions 7–9, the settlement at each monitoring point increased linearly. The growth rate in Working Conditions 7–9 was greater than in Working Conditions 1–4 (Figure 8).

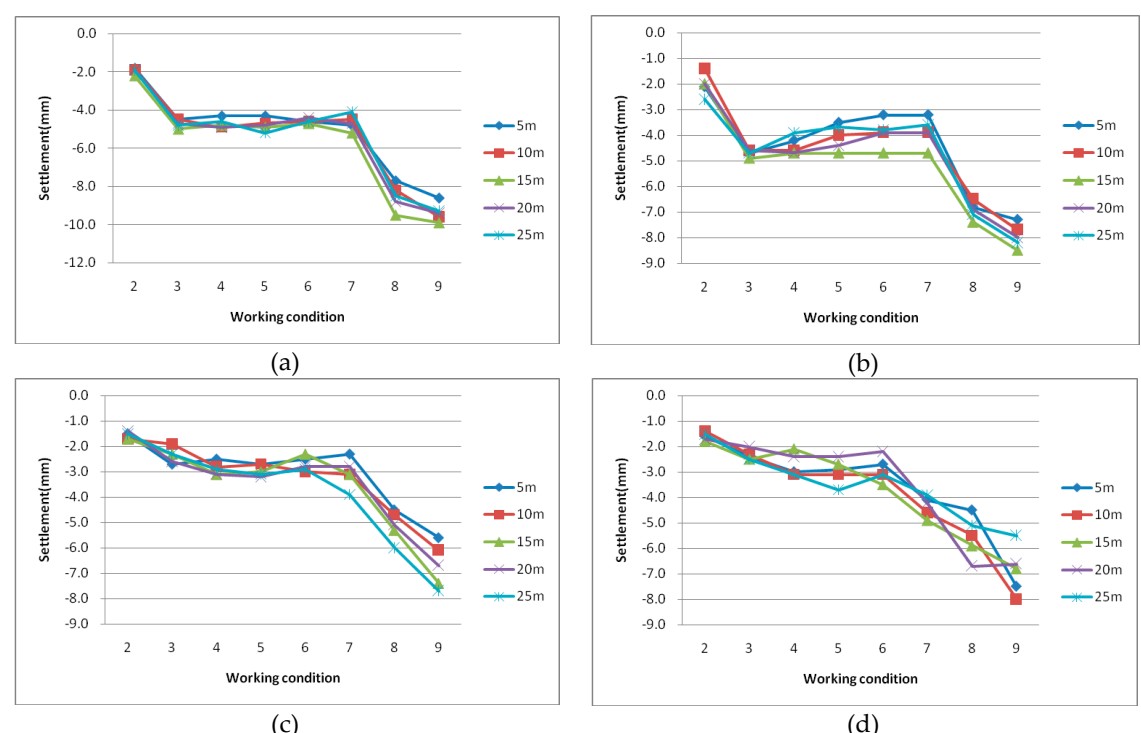

**Figure 8.** Settlement at outer edge of Slot Sections (**a**) 2, (**b**) 15, (**c**) 28, and (**d**) 42.

### 3.2. Wall-Body-Migration Analysis

Analysis of data presented in Figure 9 yielded the following results. First, the wall deformation of each working condition was linear at an excavation depth of 27 m, and the deformation curve had

segmental characteristics. The displacement of the wall body had an inflection point at a certain depth; that is, there was a peak in the displacement curve of the wall. This point gradually moved deeper with increasing depth of excavation and was generally located near the maximal excavation depth. This differed from the deformation characteristics of a cantilever pile (the lower part of the pile is fixed, and the upper part is subject to lateral thrust) because performance of the circular underground continuous wall arose from its own annular restraining force. We termed this point for each working condition the "round-underground-continuous-wall-deformation inflection point". Additionally, for Working Conditions 2 and 3, at some stage of excavation, the bottom of the wall body deviated away from the direction of the foundation pit. This was similar to the deformation characteristics of a cantilever pile. Working Conditions 4–9 did not exhibit a reverse offset at the bottom of the wall, and the final forward offset gradually increased with excavation depth. Finally, the wall-offset curve exhibited D-type distribution (Figure 9), and the maximal offset appeared at approximately 2/3 of the excavation depth. The maximal value of the inflection point was Working Condition 9, which had an offset of 6.1 mm.

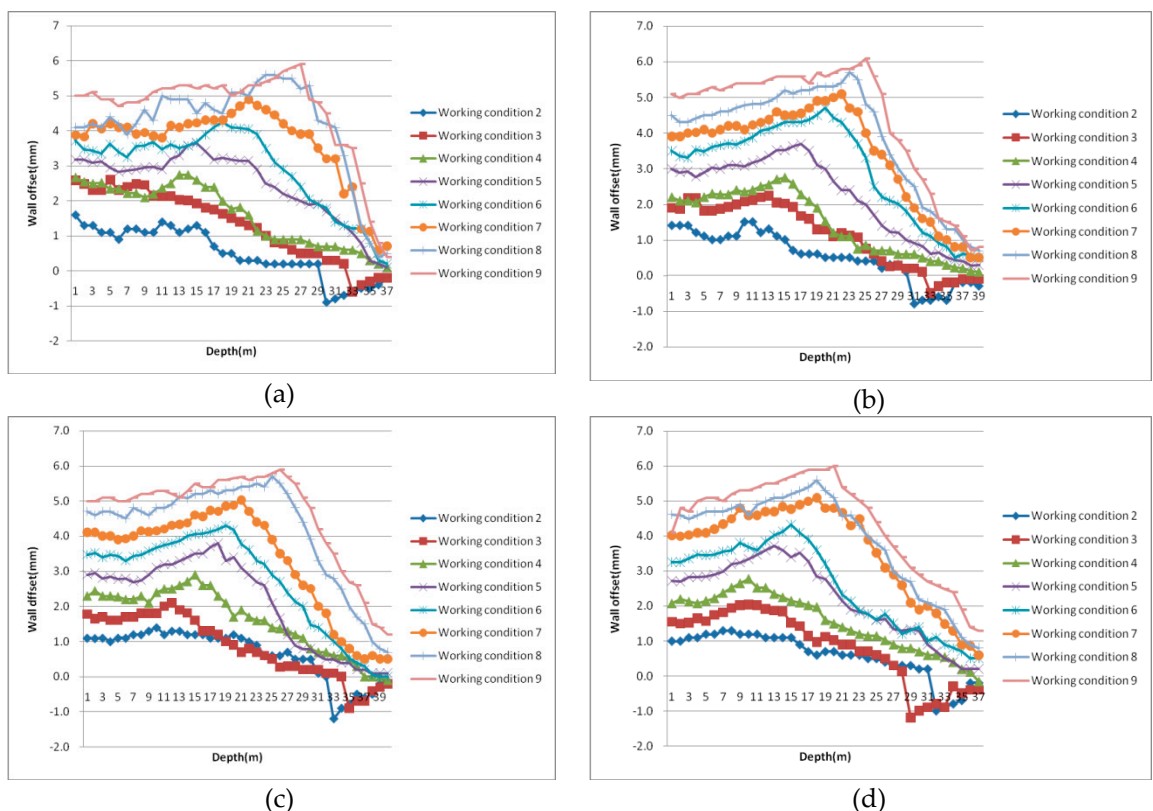

**Figure 9.** Offset around wall of Slots (**a**) 2, (**b**) 15, (**c**) 28, and (**d**) 42.

### 3.3. Settlement Analysis around Foundation Pit

There were only a few buildings and communities around the foundation pit. Thus, the construction machinery and the soil load near the foundation pit were the main factors for settlement. Settlement around the foundation pit is shown in Figure 10 for excavation Working Conditions 2–9.

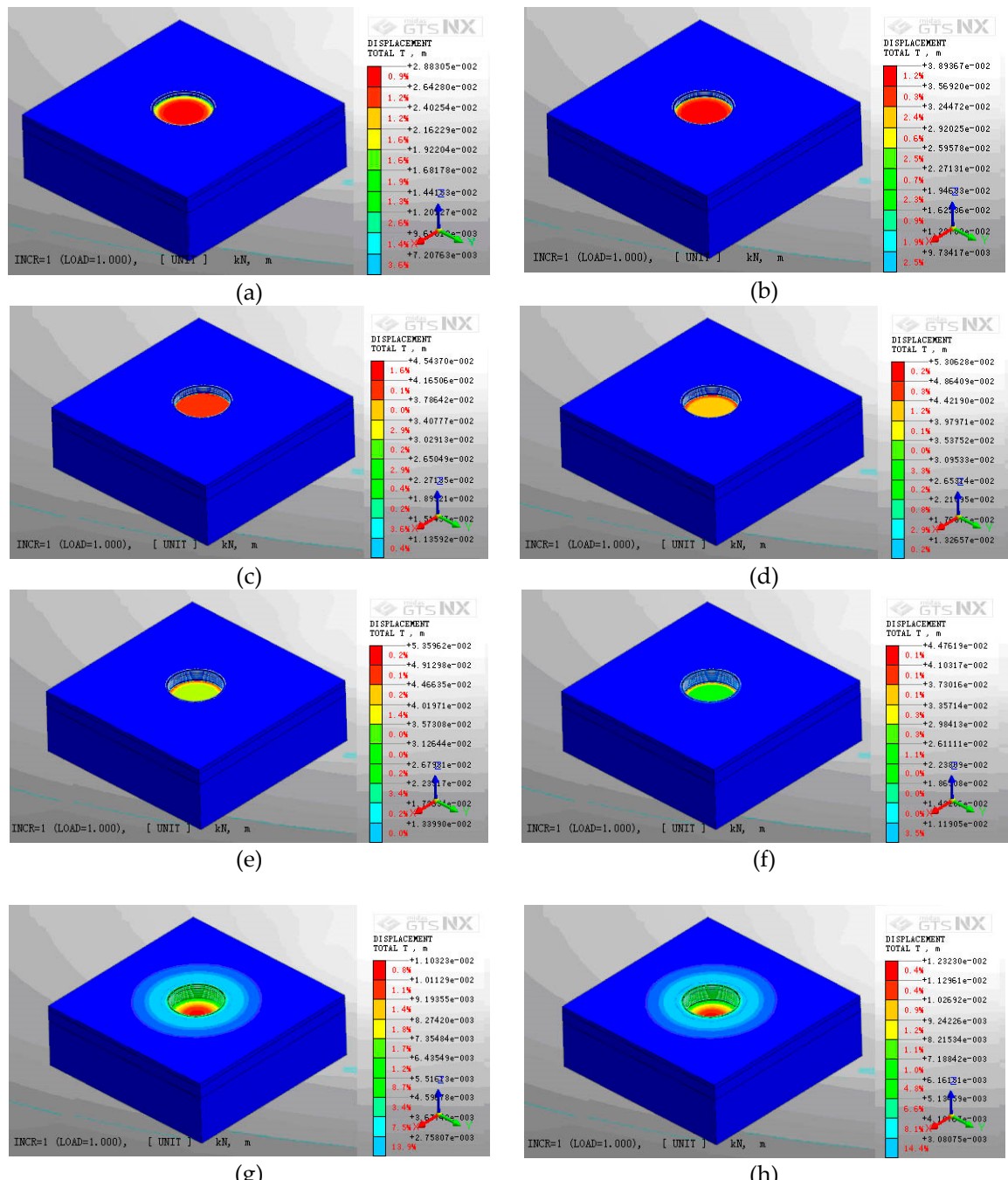

**Figure 10.** Settlement cloud around foundation pit for Cases (**a**) 2, (**b**) 3, (**c**) 4, (**d**) 5, (**e**) 6, (**f**) 7, (**g**) 8, and (**h**) 9.

Surface settlement of the outer edge of Slots 2, 15, 28, and 42 was also investigated. Due to limitations of the grid and the calculation of the model, settlement analysis was performed for 4, 8, 12, 17, 22, 27, 37, 47, 57, and 70 m depth (Figure 11).

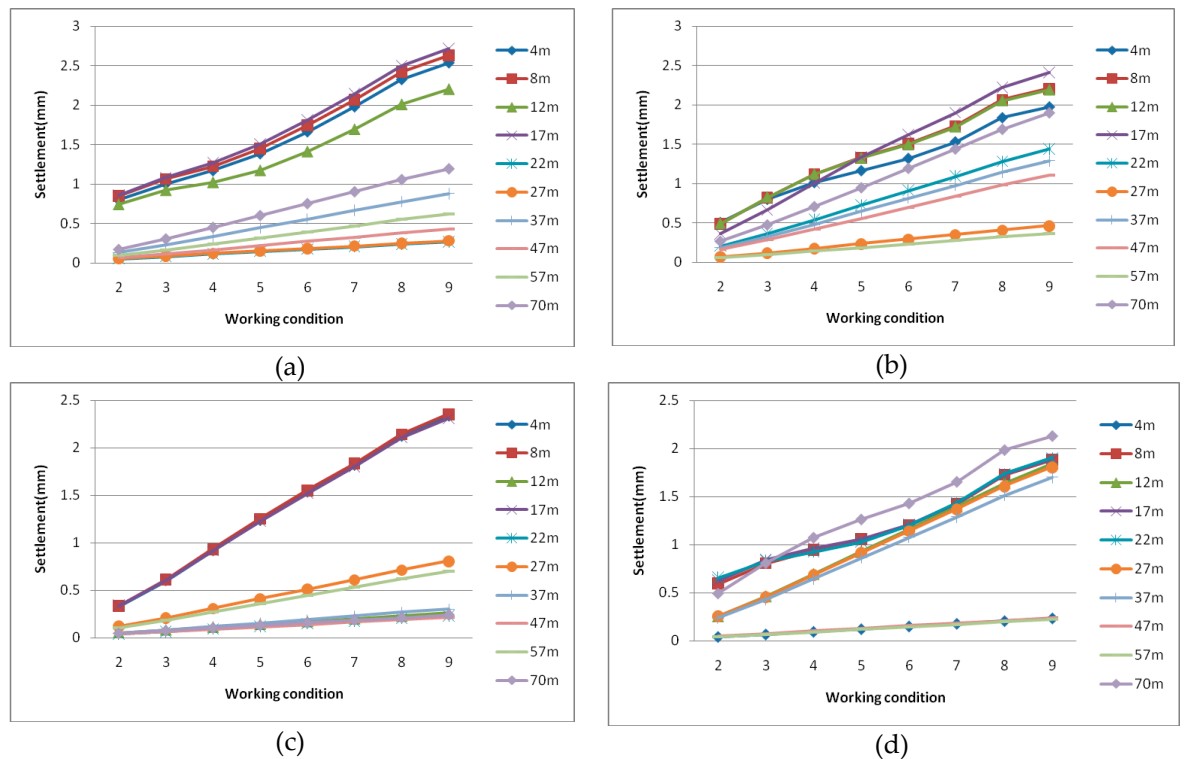

**Figure 11.** Settlement of outer edge of Slot Sections (**a**) 2, (**b**) 15, (**c**) 28, and (**d**) 42.

Figure 11 shows that surface settlement was linear and increased with the excavation depth of the foundation pit. Surface settlement within a range of about 27 m outside the foundation pit rapidly increased with the increase of excavation depth. The amount of ground settlement beyond the surface of the foundation pit, about 50 m, was slightly affected by the excavation depth of the foundation pit. Maximal surface settlement was located near the edge of the foundation pit, with a maximal value of 2.715 mm.

### 3.4. Displacement of Underground Continuous Wall

During the excavation of the foundation pit, the underground continuous wall was affected by soil stress and became offset. The wall deviation of the foundation pit for each working condition of the excavation is shown in Figure 12.

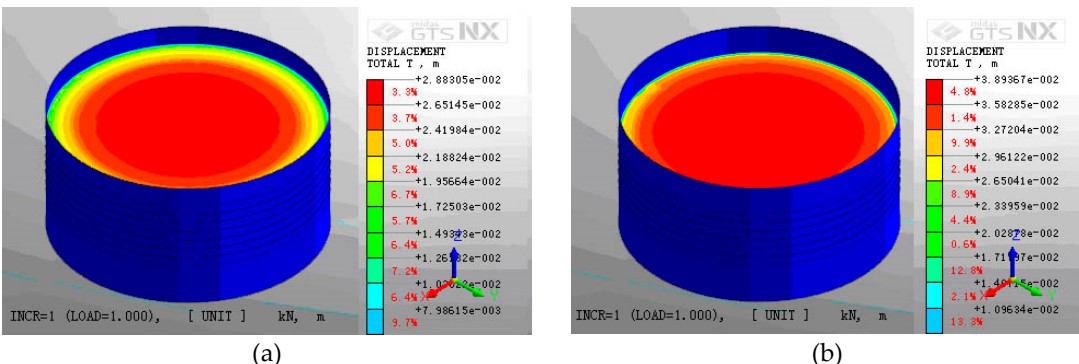

**Figure 12.** *Cont.*

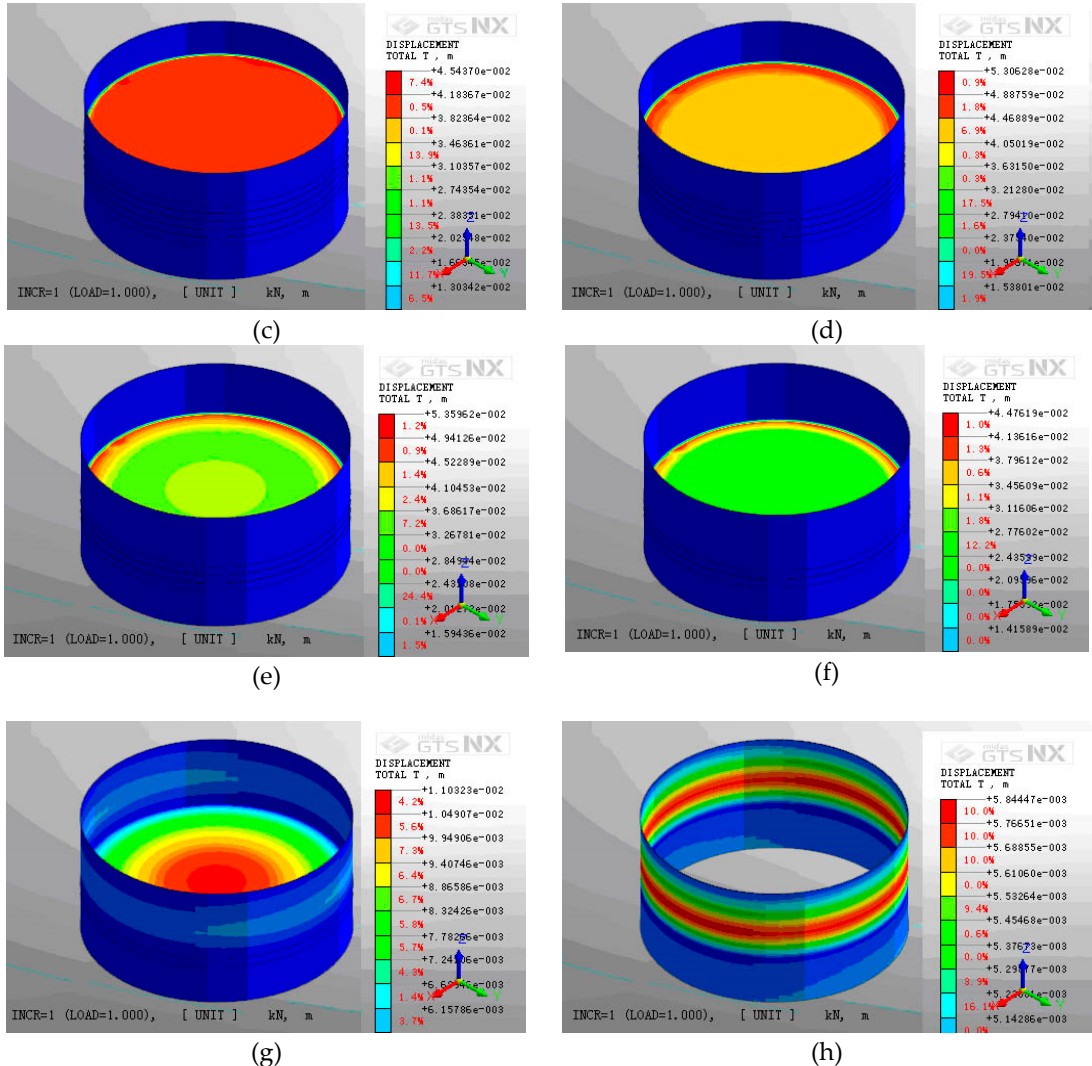

**Figure 12.** Underground diaphragm wall deviation for Cases (**a**) 2, (**b**) 3, (**c**) 4, (**d**) 5, (**e**) 6, (**f**) 7, (**g**) 8, and (**h**) 9.

The displacement model of the underground continuous wall at Slots 2, 15, 28, and 42 was selected for data processing. Due to limitations of the grid and the operation of the model, analysis of the displacement was performed for the 3, 6, 9, 12, 15, 18, 21, 24, 27, 30, and 40 m positions (Figure 13).

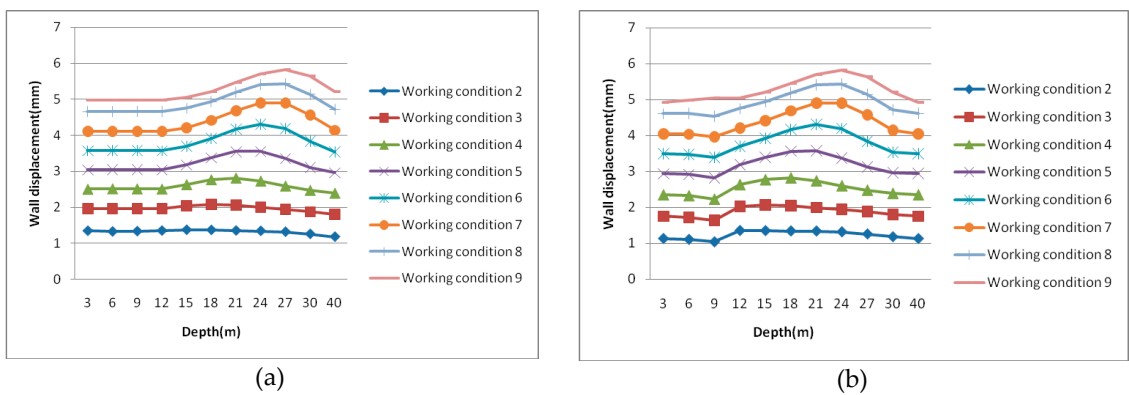

**Figure 13.** *Cont.*

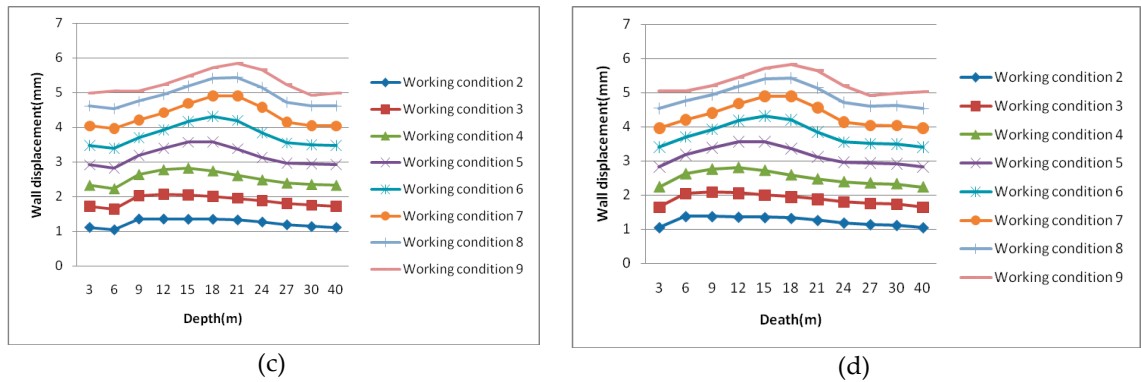

**Figure 13.** Wall offset of Slots (**a**) 2, (**b**) 15, (**c**) 28, and (**d**) 42.

These results revealed the following. First, the wall-body offset linearly increased with the depth of the excavation. Furthermore, the wall offset of each working condition showed a peak, after which the wall-body offset began to decrease. As excavation depth increased, the maximal offset of the wall shifted deeper. Second, there was no reverse offset calculation; the maximal offset of the wall was concentrated at a depth of approximately 2/3 of the total excavation depth. Third, as the depth of the excavation increased, the wall-offset curve showed D-shaped distribution. The simulated maximal offset was 5.837 mm.

Existing analysis of the deformation of the supporting wall of underground-continuous-wall foundation pits and the surrounding surface settlement mostly focuses on simple theoretical calculations [1,2,9–11] or finite-element analysis [3,4,7,8,12,13] that lack(s) validation of the used parameters. In this study, monitoring data were analyzed to identify the deformation law and other characteristics of the support structure. Three-dimensional numerical simulation of the foundation-pit excavation was conducted in Midas/GTS NX. In the process of realizing the analysis of ground settlement and wall offset around the circular underground continuous wall during construction, this paper demonstrated a key data self-verification method based on monitoring data, a breakthrough in difficulties in the selection of construction-safety calculation and finite-element-analysis parameters of foundation-pit engineering. It provides a new way of parameter selection for the sustainability study of foundation-pit and geotechnical engineering. In addition, we obtained the characteristics of surface settlement and wall offset around the circular underground continuous wall. The inflection point of the displacement of the circular underground continuous wall was proposed. These results are of great significance for the construction guidance of special-shaped underground continuous walls, providing an important reference for the continuous promotion of circular underground continuous walls.

## 4. Discussion

Monitoring and simulation results and analysis were as follows. The settlement of the surface surrounding the circular underground continuous wall was mainly affected by the depth of the foundation-pit excavation. As excavation progressed, both monitoring and simulation data showed good linearity. Monitored maximal settlement showed that the simulated value was a conservative calculation.

In addition, the deformation of the wall for each working condition showed linearity with clear staged characteristics. In particular, the deformation curve had obvious inflection points, most of which were located deeper than 2/3 of the overall excavation depth. The characteristics of the cantilever pile were not obvious in Working Conditions 3–9, but the distribution of the wall body offset in a D-shaped curve was evident. Deviation between the monitoring value of the maximal wall offset and the simulated value was only 4.31%; thus, monitoring and simulation data were in good agreement. Furthermore, force-deformation characteristics were different from those of the cantilever pile. The monitored value showed more convergence at the bottom of the wall, while analog

values were not obvious. Preliminary analysis suggests that this was because monitoring data showed increased rock-embedded rock mass at the bottom of the wall compared to the simulated data.

## 5. Conclusions

This study drew three main conclusions. First, it was determined that the surface settlement of a circular underground continuous wall is mainly controlled by the depth of foundation-pit excavation. Both monitoring and simulation data demonstrated increased linearity as excavation progressed. Appropriate physical and mechanical parameters for key data self-verification were proposed and utilized to compensate for the shortcomings of on-site monitoring data, and the extent of surface settlement caused by construction excavation was determined. Second, analysis, monitoring, and simulation results showed that the deformation of the circular underground continuous wall had unique constraint characteristics. The wall offset of each working condition showed a peak, after which the wall-body offset began to decrease. On this basis, the concept of a round-underground-continuous-wall deformation inflection point was proposed. Finally, we determined that the deformation pattern of the circular underground continuous wall showed distinct linearity, the deformation curve had an inflection point, and most of the inflection points were located below 2/3 of the excavation depth. In addition, wall-offset distribution showed an evident D-shaped curve.

The key data self-verification method proposed in this paper can be used as a method to check the validity of simulation parameters, and subsequent research can extend this method to other computing systems. This method is expected to build a bridge between monitoring data and simulation results. The concept of a round-underground-continuous-wall deformation inflection point, proposed in the paper, needs to further be applied to the quantitative relationship between the displacement of the inflection point and excavation depth.

**Author Contributions:** X.G.; conceptualization, methodology, software, validation, data analysis, investigation, resources, data curation, writing—original-draft preparation, review and editing, data visualization, and project supervision. W.-p.T.; conceptualization, validation, and funding acquisition. Z.Z.; validation and project administration. All authors have read and agreed to the published version of the manuscript.

**Funding:** This research was funded by the Western Transportation Construction Science and Technology Project (2006-318-000-07), the China Communications Construction Co., Ltd (CCCC) Technology Research and Development Project (2011-ZJKJ-01), the National Natural Science Foundation of China (51708043), and the Fundamental Research Funds for the Central Universities, CHD (300102219106).

**Acknowledgments:** We would like to thank Editage (www.editage.com) for the English language editing.

**Conflicts of Interest:** The authors declare no conflicts of interest.

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
