# Peer review of "Analysis of Deformation Characteristics of Foundation-Pit Excavation and Circular Wall"

_sustainability, doi:10.3390/su12083164_

Round 1

Reviewer 1 Report

The paper investigated the anchored, circular, underground continuous wall of the 10 Humen Second Bridge West foundation pit project. The following are some pointers to improve the paper.

Abstract

There is need to state the outcome accruing from the variables surrounding the data, model, etc, before the proposed approach at the end is stated. The authors also need to revisit this abstract in its entirety for a complete/well written abstract with the problem statement, research aim, objectives and recommendations/conclusions clearly stated. The keywords need to be in alphabetical order, usually separated by commas.

Introduction

A report is provided on previous studies done on the research topic. These studies need to be more focused by identifying and discussing respective themes. That is, provide a more focused review by giving topical alignments with the current research rather than mere reporting on what the different research were about. Then identify the gap which the current research needs to fulfil. So line 27-86 needs to be revised to precision which should end with gap identification. Then line 87-95 will identify the avenues to fill the gap which will also state the research aim or objectives. Literature review section will be better after the introductory section.

Literature review

A review of literature section needs to be provided that links up with the various sections of the paper. This should also include most of the cited work as can be seen in the introduction section. Also, a statement that sets the stage for this research is needed somewhere around line 87-95, and therefore the materials and methodology section that follows would be in the right place for smooth readership.

Materials and experimental methodology

This section seems to be well presented. However, authors need to pay attention to editing issues as in line 98 (‘pit was as follows’), line 226 (‘was repeated’) and elsewhere in the paper. Also, everything stated in this section needs to be fully explained/justified, e.g., why was the underground wall divided into 2 sections with trough sections divided into 54 sections in line 105 and not more or less than the said numbers. Was there some standard or code being followed? What was behind the sensor arrangements in line 122? Check for such instances in the paper and provide reasons accordingly. Provide annotations to the figures such as in line187, 188, etc. to explain/define them better. Soil parameters and indices (including ranges-providing their meanings) need to be explained including their implications in this project. How does the equation in line 219 operate considering this research project? Isn’t the statement on line 220-221 something to be in the result section eventhough it is a simulation? Authors need to justify why the simulation was necessary that warrant reporting it here and not in the result section.

Results section

This section is relatively well presented but needs fine tuning and more description of the figures. The figures also need to be visually clearer for smooth readership

Discussion section

This section does not seem exhaustive enough given the magnitude of the paper. Authors may need to separate discussion section from conclusion section to ensure clarity. Discussion is not a summary of results. The idea is to have the discussion section which links up with the results section and supporting (or not supporting) the items listed in the literature review section. In fact, considering this write-up, the authors may strongly consider combining the results and discussion section for a smoother readership, and then having a conclusion section. Therefore, the current paper does not provide a clear cut path for a well written research paper.  Thus, its contribution to the body of knowledge is not quite clear. Are there any areas for further research based off the discussion and conclusion?

References/citations need to be according to the format of the sustainability journal.

Overall, the paper needs improvement in many areas for smooth readership, research generalizability and publication. Edit or proofread to remove some unnecessary errors.

Reviewer 2 Report

  1. There are many typos and punctuation problems must be fixed. 
  2. Leave a blank space between the numbers and units.
  3. Avoid using "/" to separate axis titles and the units. Put the units in parentheses or use a comma to separate the units from the titles.
  4. L98: Where is the silt layer on top of the muddy soil?
  5. Figure 1: Needs units.
  6. Figure 2: Needs units.
  7. Figure 3: Text size is too small.
  8. Figure 4: Text size is too small.
  9. Figure 4: The figure is located before its description in the text.
  10. Table 1-Last Column: Change "density" to "unit weight".
  11. Table 1: Number of the digits after the decimal point is not consistent.
  12. Table 2: Change "Weight measurement" to "unit weight".
  13. L219: Add equation number.
  14. Figure 11: Move the horizontal axis numbers either to the very top or bottom.
  15. Figures 13 and 15: Text size is too small.
  16. A conclusion is expected after the discussion.
  17. References 14 through 18 haven't been used in the text and must be removed.

Reviewer 3 Report

Brief Summary of the paper entitled Analysis of the Deformation Characteristics of a Foundation Pit Excavation and a Circular Wall:

By the use of a commercial finite element code: MIDAS/GTS NX the authors perform a simulation of a foundation pit excavation in order to predict the surrounding settlement as well as continuous wall displacements. The constitutive equations used in the model are Mohr-Coulomb both in the soil and the concrete. An accurate geology profile of the site is shown and properly parametrized into the numerical model. Using ‘key data self-verification method’ the authors compare the settlement results given by the model to the available measurements in order to determine whether or not the numerical results are reasonable in case of lack of site date. Continuous wall displacements are taken as reference and an error ratio is established as follows: Deviation rate = (analog value - monitored value) / monitored value. The paper ends with a discussion  

Broad comments:

The paper is well written and structured and the English grammar and syntax is fine.

Many typos are present due to the lack of spaces between words, probably after copying the text from one text editor to another. In the line by line section those are listed.

The paper does not present any novelty concerning numerical/analytical/experimental developments. The commercial finite element code used is a black box for the potential reader and no information about the numerical implementation is available, a very simplistic Mohr-Coulomb constitutive law is used for all the materials, no theoretical expressions about this problem are provided.

The ‘key data self-verification method’ which could be the only originality of the paper is based on a deviation rate which is normalized by the monitored value. The problem comes when the monitored value is zero, which can happen in the wall inflexion point, in that case the deviation rate will be infinity. Even if this method is often used in engineering practice it cannot be used in research because of lack of soundness.

The paper thematic does not fit the aim and scope of the journal ‘Sustainability’

Line by line comments:

31 remain challenging

33 motivate?

44 for,

94 utlines à outlines?

98 was as follows

150 define phi

157 at a

158 was divided

160 simulation it

166 maximize the  simulation à simplify the simulation?

176 foundation pit. Its

183 the artificially

184 various units, final total

185 nodes. Figure 5, Figure 6 shows

  1. Figure 7 does not show calculation results
  2. Mohr-Cullen ???

206 gaps in

209 what are theoretical defects?

211 could be examined to

214 could be performed

219 dividing by zero issues?

223 for the

224 identify the

226 was repeated

226 calculation were evaluated

232 influenced by

235 or slightly

236 for working

243 for a

244 an inflection

248 a cantilever

254 curve exhibits a

261 are relatively

263 in the

266 sedimentation à settlement or subsidence

286 m deep

290 an inflection point

321 we illustrated

Round 2

Reviewer 1 Report

The abstract has improved, however, it should be revised to meet the journal’s word limit and removing the paragraphing of the abstract; may consider block type. Author’s need to revisit the requirements in its entirety including fully editing the paper. The author(s) don’t seem to have a section for the literature review as was required before. The author(s) seem to have divided the discussion and conclusion section. But how does the results and literature review correlate? This needs to be reflected in the discussion section. After reading this revised version, it is important to have the results and discussion section combined for smooth readership. The discussion section mentions ‘data was...’ when this should be ‘data were...’. Author(s) need to revise this and in the whole paper. Overall, try to be neutral and avoid use of words such as ‘our’ line 33, ‘we’ in line 397 and all others in the paper; use third person as much as possible.

Reviewer 3 Report

The reviewer appreciates the changes made by the authors. The changes improve the quality of the presentation and highlight the strengths of the paper. Many missing spaces are still present, probably due to some editing problem, this needs to be checked prior to publication.

Nonetheless I maintain my previous advice to the editor; a commercial Finite Element Model using a Mohr-Coulomb constitutive law with the ‘key data self-validation theory’ does not bring enough novelty to be considered for publication. Concerning the scope of the journal, I leave this decision up to the editor, but I do not think this paper fits the aim and scope of Sustainability.
